# Disparities in anti-SARS-CoV-2 reactivity according to vaccines administered in the era of omicron in Cameroon: Lessons for future outbreak response

Ezechiel Ngoufack Jagni Semengue[1], Desire Takou[1], Marina Potesta[2], Sandrine Claire Ndjeyep Djupsa[1], Carla Montesano[2,3], Collins Ambes Chenwi[1,4], Grace Beloumou[1], Alex Durand Nka[1,4], Aurelie Minelle Kengni Ngueko[1,4], Evariste Molimbou[1,3,4], Naomi-Karell Etame[1,5], Davy-Hyacinthe Gouissi Anguechia[1,5], Audrey Rachel Mundo Nayang[1], Pamela Patricia Tueguem[1], Therese Ndomgue[1,2], Derrick Tambe Ayuk Ngwese[1,5], Larissa Gaëlle Moko Fotso[1,5], Carlos Michel Tommo Tchouaket[1,6], Aude Christelle Ka'e[1,4], Nadine Fainguem[1], Cyrille Alain Abega Abega[1], Nadia Mandeng[7,8,9], Emilienne Epee[5,8,9], Linda Esso[8,9], Georges Etoundi Mballa[8,9], Maria Mercedes Santoro[4], Anne-Cecile Z-K Bissek[10], John Otokoye Otshudiema[11], Claudia Alteri[12], Yap Boum II[5,8,9], Anne-Geneviève Marcelin[13], Francesca Ceccherini-Silberstein[4], Alexis Ndjolo[1,5], Carlo-Federico Perno[1,14], Jean Kaseya[15], Vittorio Colizzi[1,3�], Nicaise Ndembi[15,16�]*, Joseph Fokam[1,8,17�]*

1 Chantal BIYA International Reference Centre for Research on HIV/AIDS Prevention and Management, Yaoundé, Cameroon, 2 Department of Biology, Faculty of Sciences, University of Rome "Tor Vergata", Rome, Italy, 3 Faculty of Science and Technology, Evangelic University of Cameroon, Bandjoun, Cameroon, 4 Department of Experimental Medicine, Faculty of Medicine and Surgery, University of Rome "Tor Vergata", Rome, Italy, 5 Faculty of Medicine and Biomedical Sciences, University of Yaoundé I, Yaoundé, Cameroon, 6 School of Health Sciences, Catholic University of Central Africa, Yaoundé, Cameroon, 7 Faculty of Health Sciences, University of Bamenda, Bamenda, Cameroon, 8 National Public Health Emergency Operations Coordination Centre, Ministry of Public Health, Yaoundé, Cameroon, 9 Department of Disease, Epidemic and Pandemic Control, Ministry of Public Health, Yaounde, Cameroon, 10 Division of Health Operational Research, Ministry of Public Health, Yaounde, Cameroon, 11 COVID-19 Incident Management Team, World Health Organization, Country Office – Yaoundé, Yaoundé, Cameroon, 12 Department of Oncology and Hemato-Oncology, University of Milan, Milan, Italy, 13 Virology Laboratory, University Hospital Pitie Salpetriere-Charles Foix, Paris, France, 14 Bambino Gesu' Children's Research Hospital, Rome, Italy, 15 Africa Centres for Disease Control and Prevention (Africa CDC), Addis Ababa, Ethiopia, 16 Institute of Human Virology, University of Maryland School of Medicine, College Park, Maryland, United States of America, 17 Faculty of Health Sciences, University of Buea, Buea, Cameroon

उ These authors contributed equally to this work.
* josephfokam@gmail.com (JF); NicaiseN@africa-union.org (NN)

## Abstract

With the advent of COVID-19, anti-SARS-CoV-2 vaccines were a global health priority, but evidence on their significance within tropical settings remained limited. We sought to assess the distribution of anti-SARS-CoV-2 antibodies according to vaccine status and types of vaccines administered in Cameroon during Omicron waves. A community based cross-sectional sero-survey was conducted from February-15 through July-31 2022 among individuals tested for COVID-19 in Yaoundé-Cameroon. Sociodemographic data were collected from participants. Anti-SARS-CoV-2 antibodies (both IgG and IgM) were tested on plasma and statistical analyses were performed wherever appropriate. Logistic regression was done with p<0.05 considered

**Data availability statement:** All data analyzed during the study are described within the manuscript.

**Funding:** The study was supported by the European and Developing Countries Clinical Trials Partnership (EDCTP), under the reference number EDCTP PERFECT-Study, RIA2020-EF3000. The grant was obtained by Joseph Fokam.

**Competing interests:** The authors declare no existing conflicts of interest. The funders had no role in the design of the study; in the collection, analyses, or interpretation of data; in the writing of the manuscript; or in the decision to publish the results.

statistically significant. Overall, 2449 participants were enrolled: median-age was 40 [31–49], 56.4% (1382/2449) men, 2.2% (54/2449) with flu-like symptoms and 19.6% (481/2449) reporting previous SARS-CoV-2 positivity. Regarding COVID-19 vaccination, 67.5% (1652/2449) had received at least one dose, 55.0% (909/1652) two-dose series and 37.1% (613/1652) received additional booster doses. Median duration from vaccination to phlebotomy was 5 [4–9] months. Seroprevalence of anti-SARS-CoV-2 antibodies was 81.1% (1987/2449). Following logistic regression, vaccine status (aOR=1.95), booster doses (aOR=1.36), post-vaccination time (≤5 months; aOR=1.64), Pfizer (aOR=2.07) and Moderna (aOR=1.52) vaccines, were all associated with a high prevalence of anti-SARS-CoV-2 antibodies (all p<0.05). This high seroprevalence of anti-SARS-CoV-2 antibodies suggests a certain degree of immunity/protection at community-level in Cameroon during Omicron waves, with Pfizer and Moderna inducing greater immunogenicity. However, rapid antibody waning (~5 months) calls for vaccine updates with novel variants (arising from a rapidly evolving virus) that could compromise already acquired immunity.

## Introduction

Coronavirus disease 2019 (COVID-19), caused by the severe acute respiratory syndrome coronavirus-2 (SARS-CoV-2), has spread worldwide with clinical, economic, social and even political implications [1–3]. Since the disease outbreak, several preventive measures, alongside programmatic strategies (precisely the *tracking-testing-treatment* strategy and later on genomic surveillance platforms), were implemented in many countries to limit the rapid spread and assess timely and contextual responses to COVID-19 [4,5]. Among the preventive measures that were implemented, COVID-19 vaccination remained a global health priority.

As of December 2020, more than 200 candidate vaccines against COVID-19 were in development and at least 52 of these reached the stage of clinical trials involving humans [6]. There are three main methods in vaccine development. Their differences lie in whether they use a whole virus (attenuated or inactivated); only viral particles that will trigger the immune system; or only the genetic material that provides the instructions for making specific proteins and not the whole virus [6]. Vaccine technologies developed for COVID-19 used next generation strategies for precise targeting of COVID-19 infection mechanisms, among which we found nucleic acid technologies (nucleoside-modified messenger RNA and DNA), non-replicating viral vectors, peptides, recombinant proteins, live attenuated viruses, and inactivated viruses [7,8]. Even though several reports supported the necessity of vaccine introduction in high income countries, Africa in particular was very slow and hesitant towards COVID-19 vaccination [2,9–11]. In effect, the multiplicity of false information as well as the low vaccine manufacturing capacity on the continent were always cited as the main reasons for this hesitancy; and this led to numerous gaps in the overall pandemic preparedness and disease coordination [2,12–14]. In the meantime, the molecular epidemiology of SARS-CoV-2 was characterized by a rapid viral evolution and translated by the emergence of many different variants worldwide. A key determinant of the rate at which a SARS-CoV-2 evolved was its mutation rate. This is the intrinsic rate at which genetic changes occur per replication cycle and that of SARS-CoV-2 was estimated at around $1\times10^6 – 2\times10^6$ mutations per nucleotide per replication cycle; in line with what has been described for other betacoronaviruses [15]. The emergence of

SARS-CoV-2 variants that posed an increased risk to global public health prompted the World Health Organization (WHO) to characterize some as variants of interest (VOIs), variants under monitoring (VUM) and variants of concern (VOCs) in order to prioritize global monitoring and research, and to inform and adjust the COVID-19 response. Among the latter, four major VOCs were genetically distant from the original lineage and divided the pandemic into four epidemiological waves; these were the "alpha" variant (B.1.1.7), the "beta" variant (B.1.351), the "delta" variant (B.1.617.2) and the "omicron" variant (B.1.1.529) (https://covdb.stanford.edu/variants/).

In 2021, omicron variant of SARS-CoV-2 was reported to the WHO and the Africa Centres for Disease Control and Prevention (Africa CDC) by the Network for Genomics Surveillance in South Africa [16,17]. It was first detected in Botswana and has rapidly become the predominant variant in circulation worldwide, with different sub-variants that have emerged from the original omicron lineage; and this was mainly attributable to its unprecedented mutational rate [15,16,18]. Notably, the particularly high number of mutations of omicron both in the receptor binding domain of its spike protein and other key proteins could help to justify the rapid evolution of this VOC and further explain the peculiarity of its features. Unlike the delta-variant, the omicron wave was characterized by a rapid transmissibility, but fewer patients were admitted to hospital, less severe illness, and a lower case-fatality rate [16–19]. However, the high mutational rate of SARS-CoV-2 and specifically of omicron variants led to questioning about the preservation of vaccine efficacy and, to a certain extent, the level of immune-escape of these omicron sub-variants [10,18,20].

In Cameroon, like many other low and middle income countries (LMICs) in Africa, the main COVID-19 vaccines administered were Astra-Zeneca (ChAdOx1 nCoV-19), Sinopharm (BBIBP-CorV), Janssen (Johnson & Johnson or Ad26.COV2.S), Moderna (mRNA-1273), Pfizer-BioNTech (COMIRNATY or BNT162b2) and Sputnik-light (Gam-COVID-Vac) [21]. During the omicron wave, vaccine hesitancy significantly decreased despite the gap in achieving the national target for adequate vaccine coverage. With few cases of breakthrough infections reported locally and a lack of contextual data supporting vaccine effectiveness, we sought to assess the overall prevalence of anti-SARS-CoV-2 antibodies, its disparity according to vaccine status and types of vaccines administered in Cameroon. In effect, SARS-CoV-2 serological antibodies just like any other antibodies in the immune system will help to protect against future infections by the same pathogen and are specifically essential to appreciate the true extend of disease penetration within the community; thus providing orientations for disease control. Therefore, in order to achieve our study objective, we have employed two rapid serological tests for COVID-19 antibodies among those widely available nationwide: the Ninonasal COVID-19 IgG/IgM all in one test which detects antibodies against the SARS-CoV-2 nucleocapsid protein [22]; and the ABBEXA COVID-19 IgG/IgM device captures antibodies against both SARS-CoV-2 spike and nucleocapsid proteins [23].

## Materials and methods

### Ethics statement

The study obtained ethical approval from the Cameroon National Ethics Committee for Human Health Research (reference N°2022/01/1430/CNERSH/SP). Following the Helsinki's declaration and the national regulations, informed consent was obtained from all participants. Participants who did not provide a written informed consent were not considered for inclusion in the study. The use of specific identifiers ensured confidentiality; and data were protected by using a password-protected computer.

## Study design

This was a community based cross-sectional design in which consenting participants were recruited from the 15th of February to the 31st of July 2022 in the city of Yaoundé, Cameroon. This recruitment period corresponds to the active phase of Omicron circulation locally (end-2021 throughout 2022).

## Study site

The study was hosted at the Chantal BIYA International Reference Centre for research on HIV/AIDS prevention and management (CIRCB). CIRCB is committed to research on HIV/AIDS prevention and management and it is a governmental institution under the Cameroonian Ministry of Public Health (http://www.circb.cm/btc_circb/web/). Importantly, CIRCB was designated as one of the COVID-19 reference centers by the Cameroonian Ministry of Public Health. As a reference center for COVID-19 the principal missions of CIRCB were as follows: (a) the diagnostic of SARS-CoV-2 infection through antigen rapid diagnostic tests (Ag RDT) for symptomatic and asymptomatic individuals, and through real-time reverse transcriptase polymerase chain reaction (rt-RT PCR) both for suspected cases and international travelers; (b) serological surveys to assess the level of antibodies in the community; (c) and the genomic surveillance of SARS-CoV-2 variants and sub-variants circulating at country-level for a better coordination of the response against the disease progression. Only participants fully complying to the study were finally included (i.e., those consenting to participate; of both genders; aged >18years; residing in Cameroon; with available clinical data regarding previous infection and vaccination for those vaccinated).

## Participants' enrolment and data collection

All individuals attending the facility for COVID-19 diagnosis were systematically approached for potential inclusion in the study. These were either volunteers, contact cases, symptomatic cases or asymptomatic travelers. After obtaining their informed consent, socio-demographic and detailed clinical data were collected through a questionnaire which was then encoded within a database. Specifically, socio-demographic data included the name of the participants, the field code, the age, the gender, the phone number, the residence and for women we also asked if they were pregnant at the moment of the study. As clinical data, we checked for the presence of any symptoms, the type of symptoms, the presence of co-morbidities, the type of co-morbidities, previous SARS-CoV-2 infection, the vaccine status, the name of vaccine taken, number of doses and the date of the last dose of vaccination. Participants were then prepared and blood samples were collected in an ethylenediamine tetra-acetic acid (EDTA) tube. Plasma aliquots were separated later for anti-SARS-CoV-2 antibodies' detection. Nasopharyngeal swabs were also collected in a 1 ml tube containing viral transport medium for real time RT-PCR.

## Laboratory analyses

**Antibodies' detection.** Plasma aliquots of 1 ml each were constituted from blood samples by centrifugation at 1800 rpm for 15 min and stored at -20°C for a maximum of two weeks before the manipulation. Anti-Sars-CoV-2 antibodies were tested on these plasma aliquots using Ninonasal COVID-19 IgG/IgM all in one test (https://ngtest-covid-19.com/en/ng-test-igm-and-igg-all-in-one/) and ABBEXA COVID-19 IgG/IgM (https://www.abbexa.com/covid-19-products) assays, as per the manufacturers' protocols. Briefly, plasma samples were mixed by low-speed vortex after which 10 µL of supernatant was applied to the device as per

the manufacturer's instructions. A valid result consisted of the appearance of a red line in the Control (C) area of the reading window. A negative result consisted of the presence of the red line only in the C area, whereas the presence of a red line on both C and G areas indicated a reactivity to immunoglobulin G (IgG). A red line on both C and M areas indicated a reactivity to immunoglobulin M (IgM) and a red line in C and both G and M areas of the reading window indicated a positive result for both IgG and IgM.

As for other lateral flow immunoassays, both rapid tests were highly reliable for the detection of anti-SARS-CoV-2 antibodies with good intrinsic features. The principal difference between both tests was that Ninonasal COVID-19 IgG/IgM all in one test only captures antibodies against the SARS-CoV-2 nucleocapsid whereas ABBEXA COVID-19 IgG/IgM captures antibodies against spike and nucleocapsid proteins. Of note, specificity of the kits used were >98.6% and sensitivity equal to 100% [22,23].

**Nucleic acid extraction and rt-RT PCR.** Nasopharyngeal swabs were collected from the nasopharynx by trained personnel in a 1 mL tube containing standard viral transport medium. The procedure for SARS-CoV-2 real-time RT-PCR (rt-RT PCR) was as follows: viral RNA was first manually extracted from 140μL nasopharyngeal swab using the QIAamp Viral RNA Mini Kit (Qiagen Inc, Valencia, CA, USA) as per manufacturer's instructions. Amplification was then performed using the DaAn gene RT-PCR kit (https://en.daangene.com/products/covid-19-test-kits) on the Quant Studio 5 (Thermofisher) in a reaction containing for each sample, 17μl of PCR reaction mix A and 03μl of PCR enzyme mix B provided with the kit. The rt-RT-PCR conditions consisted of an initial step of 1 cycle at 50°C for 15 min for reverse transcription and 1 cycle of 95°C for 15 min for *taq pol* activation. We then have for 45 cycles: 94°C for 14 sec for denaturation and 55°C for 45 sec for annealing and synthesis. The protocol used probes targeting the open reading frame (ORF1ab) gene and the nucleocapsid (N) protein gene, with a lower limit of detection of 500copies/mL and an amplification reaction of 45 cycles. As per the manufacturer's instructions, each sample with a cycle threshold (CT) value <40 was considered positive, while a CT value >40 or "undetermined" (no amplification after 45 cycles) was considered negative. We also evaluated the positivity at a CT value <37 as per our national guidelines in Cameroon. In effect, following our national guidelines, results with CT values <25 were classified as very high positives whereas those with CT values between 25–33 were moderate and CT values between 33–37 were low positives.

## Data analyses

Data collected were entered into Microsoft Excel 2021. Descriptive statistics used proportions, percentages, medians and interquartile ranges (IQR). Association analyses were performed using Pearson's Chi-square or Fisher's exact tests wherever appropriate, with all p-value <0.05 considered statistically significant. All variables with p<0.2 during bivariate analyses were retained for multivariable analysis. Before multivariate logistic regression was performed, data were checked for multicollinearity and patients with missing data were excluded from the study.

## Results

### Socio-demographic and clinical features of the participants

A total of 2449 participants were enrolled; the median [IQR] age was 40 [31–49] years, and 56.4% (1382/2449) of the participants were men. Only 2.2% (54/2449) of the participants were symptomatic and 9.7% (238/2449) presented co-morbidities at enrolment. Among symptomatic participants, the most prevailing symptoms were cough (48.1%, 26/54); running nose

**Table 1. Socio-demographic and clinical features of participants at enrolment (N = 2449).**

| Variables | n (%) |
|---|---|
| Gender | |
| Male | 1382 (56.4) |
| Female | 1067 (43.6) |
| Pregnant women | 13 (1.2)* |
| Median Age [IQR] | 40 [31–49] |
| Presence of flu-like symptoms | |
| No | 2395 (97.8) |
| Yes | 54 (2.2) |
| Various symptoms (n=54) | |
| Cough | 26 (48.1)** |
| Running nose | 18 (33.3)** |
| Headaches | 11 (20.4)** |
| Throat-aches | 10 (18.5)** |
| Shortness of breath | 9 (16.7)** |
| Fever | 8 (14.8)** |
| Loss of odor | 6 (11.1)** |
| Diarrhea | 6 (11.1)** |
| Shivering | 2 (3.7)** |
| Rashes | 2 (3.7)** |
| Presence of co-morbidities | |
| No | 2211 (90.3) |
| Yes | 238 (9.7) |
| Various Co-morbidities (n=238) | |
| High blood pressure | 118 (49.6)*** |
| Obesity | 70 (29.4)*** |
| Diabetes | 57 (23.9)*** |
| Heart injuries | 20 (8.4)*** |
| Pulmonary diseases | 14 (5.9)*** |
| Cancer | 10 (4.2)*** |
| HIV | 8 (3.4)*** |
| Sickle cells | 4 (1.7)*** |
| Asthma | 1 (0.4)*** |
| Renal diseases | 1 (0.4)*** |

IQR = interquartile range; All percentages were calculated with N = 2449 except for (*) pregnant women (n = 1067); (**) various symptoms (n = 54); and (***) various co-morbidities (n = 238).

(33.3%, 18/54) and headaches (20.4%, 11/54) whereas among those with co-morbidities, the most prevailing co-morbidities were high blood pressure (49.6%, 118/238); obesity (29.4%, 70/238) and diabetes (23.9%, 57/238) (see Table 1 for detailed socio-demographic and clinical parameters at enrolment).

## SARS-CoV-2 infection and COVID-19 vaccination

About 19.6% (481/2449) of the participants declared previous SARS-CoV-2 positivity through PCR. At enrolment, SARS-CoV-2 positivity was 8.5% (208/2449) as per the manufacturer's threshold (CT <40) and 3.4% (84/2449) at the national positivity threshold (CT

<37). Specifically, positivity was 3.70% (2/54) among symptomatic participants versus 3.42% (82/2395) among the asymptomatic (p=0.71). According to viral load, 1.2% (1/84) participant had a very high positivity, 36.9% (31/84) participants had moderate positivity and 64.2% (52/84) had high positivity.

Regarding COVID-19 vaccination, 67.5% (1652/2449) of the study population had received at least one dose of vaccine. Among those vaccinated, 66.9% (1105/1652) received mRNA vaccines i.e. 48.7% (805/1652) Pfizer and 18.2% (300/1652) Moderna. As for viral vector vaccines, 24.8% (410/1652) received Johnson & Johnson; 8.1% (134/1652) AstraZeneca and 0.2% (4/1652) Sputnik-light. Only 4.8% (79/1652) received Sinopharm, which is an inactivated virus vaccine. As for vaccine doses among these participants, 7.4% (123/1652) were partially vaccinated (i.e. who did not receive the number of doses recommended by the manufacturers for effective protection), 55.0% (909/1652) received two-dose series and 37.1% (613/1652) received additional booster doses. Median duration from the most recent dose of vaccine to phlebotomy was 5 [4–9] months. Additionally, among participants with previous SARS-CoV-2 infection, 80.9% (389/481) were vaccinated and 19.1% (92/481) were unvaccinated. Among the vaccinated with previous SARS-CoV-2 positivity, we did not have the time between infection and vaccination; but the median duration from the last dose of vaccination to phlebotomy was 6 [5–9] months.

Of note, SARS-CoV-2 positivity at the time of enrolment was 5.6% (45/797) among unvaccinated versus 2.6% (43/1652) among vaccinated participants (p=0.0003; OR=2.2 [95% Confidence Interval: 1.5–3.4]). Among the vaccinated specifically, SARS-CoV-2 positivity was 2.2% (18/805), 4.6% (19/410), 0.3% (1/300), 2.9% (4/134), 3.8% (3/79) and 0.0% (0/4) among those who received Pfizer, Johnson & Johnson, Moderna, AstraZeneca, Sinopharm and Sputnik-light respectively. About 25% (21/84) of the participants with on-going SARS-CoV-2 infection received their vaccines ≤5 months prior the start of the study; Specifically, 52.4% (11/21) had received Johnson & Johnson, 42.8% (9/21) had received Pfizer and 4.8% (1/21) had received Moderna.

## Anti SARS-CoV-2 reactivity and its determinants

Overall, the prevalence of anti-SARS-CoV-2 antibodies was 81.1% (1987/2449), with 1.2% IgM, 73.9% IgG and 6.5% IgM/IgG. Tables 2–4 below presents the proportion of anti-SARS-CoV-2 antibodies according to socio-demographic and clinical parameters, previous SARS-CoV-2 positivity, on-going infection and COVID-19 vaccination.

On univariate analysis, factors associated to a high prevalence of anti-SARS-CoV-2 antibodies were vaccine status (1398/1652 vs 589/797; OR=1.9 [95%CI: 1.6–2.4]; p<0.0001) and current infection which was inversely proportional (61/84 vs 1926/2365; OR = 0.6 [0.37–0.98]; p=0.047) (see Table 2 for more details). Among the vaccinated, those who received booster doses had higher odds for anti-SARS-CoV-2 antibodies (OR=2.5 [1.9–3.4]; p<0.0001) than those with two dose series (OR = 1.7 [1.3–2.1]; p<0.0001) and those who received partial dose (OR=1.7 [1.0–2.8]; p=0.04) when compared to unvaccinated participants. As for the duration, a post-vaccination time ≤5 months (OR=2.6 [2.0–3.4] and p<0.0001) was found as a determinant of high anti-SARS-CoV-2 antibodies. Regarding the vaccines, Pfizer, Moderna and Johnson & Johnson induced greater immunogenicity (OR=2.4 [1.8–3.1], p<0.0001; OR=2.0 [1.4–2.9], p<0.0001 and OR=1.4 [1.0–1.9], p=0.025 respectively) than Sinopharm, AstraZeneca and Sputnik-light (OR=1.8 [0.9–3.3], p=0.09; OR=1.5 [0.9–2.4], p=0.103; and OR=1.1 [0.1–10.2]; p=1 respectively) when recipients of each vaccine were compared to unvaccinated individuals (see Table 3 for further details).

On multivariate logistic regressions, the vaccination status, booster doses, post-vaccination time ≤5months, Pfizer and Moderna vaccines remained statistically associated with the high

**Table 2. Factors associated to the presence of anti SARS-CoV-2 antibodies in the study population: General distribution of anti-SARS-CoV-2 antibodies.**

| Variables | | Anti-SARS-CoV-2 antibodies | | | p-value | OR [95% CI] |
|---|---|---|---|---|---|---|
| | | Yes | No | % | | |
| Gender | Male | 1115 | 267 | 80.7 | 0.5 | 0.9 [0.8–1.1] |
| | Female | 872 | 195 | 81.7 | | |
| Age | ≤40 years | 1044 | 251 | 80.6 | 0.5 | 0.9 [0.8–1.1] |
| | >40 years | 933 | 210 | 81.6 | | |
| Symptoms | Yes | 45 | 9 | 83.3 | 0.9 | 1.2 [0.6–2.4] |
| | No | 1942 | 453 | 81.1 | | |
| Co-morbidities | Yes | 187 | 51 | 78.6 | 0.3 | 0.8 [0.6–1.2] |
| | No | 1800 | 411 | 81.4 | | |
| Previous positivity | Yes | 397 | 84 | 82.5 | 0.4 | 1.1 [0.9–1.5] |
| | No | 1590 | 378 | 80.8 | | |
| On-going Infection *(CT<37) | Yes | 61 | 23 | 72.6 | **0.047** | **0.6 [0.37–0.98]** |
| | No | 1926 | 439 | 81.4 | | |
| Vaccine-status | Yes | 1398 | 254 | 84.6 | **<0.0001** | **1.9 [1.6–2.4]** |
| | No | 589 | 208 | 73.9 | | |

OR= odd ratio; CT= cycle threshold; 95CI= 95% confidence intervals; (*) current infection was assessed according to our national CT threshold and not the manufacturer's threshold. P-values and OR in bold are those of variables associated with high prevalence of anti-SARS-CoV-2 antibodies.

**Table 3. Factors associated to the presence of anti SARS-CoV-2 antibodies in the study population: Anti-SARS-CoV-2 antibodies among vaccinated individuals as compared to unvaccinated.**

| Variables | | Anti-SARS-CoV-2 antibodies | | p-value | OR [95% CI] |
|---|---|---|---|---|---|
| | | Yes | No | | |
| Type of Vaccination | None* | 589 | 208 | 1 | / |
| | Partial | 102 | 21 | **0.04** | **1.7 [1.0–2.8]** |
| | Two-dose series | 752 | 157 | **<0.0001** | **1.7 [1.3–2.1]** |
| | Boost doses | 538 | 75 | **<0.0001** | **2.5 [1.9–3.4]** |
| Post Vaccination time | ≤5 months | 703 | 94 | **<0.0001** | **2.6 [2.0–3.4]** |
| | >5 months | 647 | 149 | **0.0005** | **1.5 [1.2–1.9]** |
| Type of vaccines | Pfizer | 701 | 103 | **<0.0001** | **2.4 [1.8–3.1]** |
| | J&J | 319 | 80 | **0.025** | **1.4 [1.0–1.9]** |
| | Moderna | 256 | 44 | **<0.0001** | **2.0 [1.4–2.9]** |
| | AstraZeneca | 103 | 24 | 0.103 | 1.5 [0.9–2.4] |
| | Sinopharm | 65 | 13 | 0.09 | 1.8 [0.9–3.3] |
| | Sputnik-light | 3 | 1 | 1 | 1.1 [0.1–10.2] |

OR=odd ratio; 95CI=95% confidence intervals; (*) All calculations in this sub-table were done considering unvaccinated participants as the reference. P-values and OR in bold are those of variables associated with high prevalence of anti-SARS-CoV-2 antibodies.

prevalence of COVID-19 antibodies (aOR=1.95, p<0.0001; aOR=1.36, p=0.046; aOR=1.64, p=0.0006; aOR=2.07, p<0.0001; and aOR=1.52, p=0.017 respectively – see Table 4 for further details).

## Discussion

The advent of the COVID-19 towards the end of 2019, and its subsequent fast spread worldwide was unprecedented. For the past four years, anti-SARS-CoV-2 vaccines were considered

**Table 4. Factors associated to the presence of anti SARS-CoV-2 antibodies in the study population: Determinants of high prevalence of anti-SARS-CoV-2 antibodies.**

| Variables | | aOR [95% CI] | p-value |
|---|---|---|---|
| On-going Infection | (Yes/No) | 0.67 [0.41–1.11] | 0.119 |
| Vaccine-status | (Yes/No) | **1.95 [1.57–2.43]** | **<0.0001** |
| Type of Vaccination | Partial/Complete | 1.17 [0.69–1.96] | 0.564 |
| | Boost-doses/Complete | **1.36 [1.0–1.85]** | **0.046** |
| Post Vaccination time | (≤5 months/ >5 months) | **1.64 [1.24–2.18]** | **0.0006** |
| Type of vaccines | Pfizer | **2.07 [1.61–2.65]** | **<0.0001** |
| | J&J | 1.31 [0.98–1.74] | 0.064 |
| | Moderna | **1.52 [1.08–2.15]** | **0.017** |
| | AstraZeneca | 1.28 [0.81–2.04] | 0.291 |
| | Sinopharm | 1.46 [0.81–2.66] | 0.209 |

aOR= adjusted odd ratio; 95CI= 95% confidence intervals; P-values and OR in bold are those of variables associated with high prevalence of anti-SARS-CoV-2 antibodies. All results presented in the table are derived from a multiple regression analysis, considering variables with p<0.2 in Tables 2 and 3.

as a global health priority, with studies focusing on vaccines' efficacy guaranteeing the effectiveness of the response strategy worldwide. However, evidence on the clinical relevance of COVID-19 vaccination within tropical settings like Cameroon remained limited. The principal aim of this study was to inform on the effectiveness of COVID-19 vaccines in Cameroon in order to generate contextual evidence for revised strategies and a better preparedness in the advent of future outbreaks. Importantly, as already stated elsewhere, it is worth recalling that SARS-CoV-2 serological antibodies tests are essential to appreciate the true extend of disease penetration within the community. In the context of COVID-19 vaccination in Cameroon, we herein present a thorough assessment of the antibody reactivity according to vaccines administered locally.

Our first observation was the slight predominance of men in the study population, with very few participants presenting COVID-19 co-morbidities and even fewer symptomatic cases as reported worldwide during the omicron wave [16,24]. In effect, many mutations present in omicron variants and sub-variants were rarely detected in previously identified variants, particularly in the spike glycoprotein [16,24]. Such mutations, mostly residing in the receptor binding domain, play a pivotal role in enhancing SARS-CoV-2 infectivity by increasing binding affinity for angiotensin converting enzyme 2 (ACE2) mostly present among men [16,24]. Fortunately, this important mutational rate observed in omicron led to high transmissibility but less hospitalization, less severe illness and a lower case-fatality rate [16–19].

Interestingly, about one-fifth of the participants declared previous SARS-CoV-2 infection and thus had produced anti-SARS-CoV-2 antibodies from the natural infection. At enrolment, more than half of the study population had received COVID-19 vaccines; including some of those who have had antibodies from previous natural infection. Notably, the vaccine coverage observed in this sub-population of Cameroonians results from local governmental actions and restrictive measures implemented at the borders for international travelers [4,5]. This is all the more relevant as SARS-CoV-2 positivity at enrolment was found to be more than two times higher among unvaccinated participants; illustrating to a certain extend the protection against SARS-CoV-2 infection among those vaccinated [10,25–27]. Notwithstanding, the proportion of vaccinated individuals found with SARS-CoV-2 positivity was probably related to the active circulation of omicron VOC itself. In effect, epidemiological evidence from previous reports have shown that the omicron VOC

is characterized by a 2- to 3-fold increased risk of reinfection [16]. This implies that this specific SARS-CoV-2 VOC is fully capable of evading preexisting immunity (from natural infection or vaccine-mediated).

Additionally, vaccine status was found to be the first determinant of high anti-SARS-CoV-2 seropositivity; translating important archiving of antibodies (mainly IgG) following vaccination rather than natural infection [10,28]. These findings therefore demonstrate the importance of vaccine-mediated immune response; in line with the global advocacy for COVID-19 vaccination especially for low- and middle-income countries (LMICs) and underserved populations as promoted by the Africa Centres for Diseases Control and Prevention (Africa CDC) [2,10,12]. Furthermore, greater odds of anti-SARS-CoV-2 antibodies were found among vaccinated participants who received booster doses and those whose last dose was less than six months prior the start of the study. Several studies already reported the importance of booster doses as well as the rapid waning of vaccine-mediated anti-SARS-CoV-2 antibodies [10,28–30]. Moreover, the protective effect of COVID-19 vaccines is more prominent with additional doses, as Vitiello *et al.,* in 2022 already stressed on the need for booster doses and the necessity to investigate on the duration of the protection with this so called "booster vaccination" [18]. Because of waning immunity, individuals fully vaccinated and thus resistant to reinfection will typically be fewer than those with partial immunity, so intrinsically transmissible viral lineages can maintain high fitness even without immune escape in non-naive populations [15]. In the meantime, it is worth noting that protection from the combination of both natural infection and vaccination, termed "hybrid immunity", has been shown to be greater in magnitude and durability than that provided by either vaccine immunity or natural immunity alone [31,32]. Specifically, infection-acquired immunity boosted with vaccination was reported to remain high more than 1 year after infection [10]. In effect, SARS-CoV-2 natural infection has been associated to a much broader humoral and cellular immune response against spike, membrane and nucleocapsid proteins whereas vaccine-mediated immune response is known to be solely against antigens from the spike protein [31]. The combination of both protections leads to increased neutralizing antibody responses as well as T-cells responses against the spike and other proteins [31]. Thus, the rapid waning of anti-SARS-CoV-2 antibodies observed in this study could indicate the low-level of hybrid immunity in the study population (just one-fifth of the study population with previous SARS-CoV-2 infection).

Regarding the vaccines administered, only Pfizer and Moderna were found to be immunogenic at the end of the analyses. Interestingly, these were the only mRNA-based vaccines. Even though there is an important waning of immunity as already discussed, it is noteworthy that mRNA vaccines have been associated with a higher magnitude of neutralizing antibody titer directly after vaccination [10,31,32]. Unlike methods that use a whole weakened or dead virus or viral particles, mRNA vaccines use mRNA created in a laboratory to teach our cells how to make a protein - or even just a piece of a protein - that triggers an immune response inside our bodies [6,8]. The mRNA from the vaccines is then broken down within a few days after vaccination and discarded from the body [6,8]. Our results therefore advocate for a prioritization of these mRNA vaccines over other vaccine technologies in our setting, as there might be a much lower immune escape following vaccination with Pfizer and Moderna even in the era of active omicron circulation [31,32]. This is in line with previous reports highlighting higher immune evasion from other vaccine technologies especially during infection with omicron variants and sub-variants [18]. In the same line, Markov et al., have explained that many of the major VOC mutations in the spike protein are found in the receptor binding domain and amino terminal domain where neutralization antibody binding is the most potent [15].

To our knowledge, this is the first study evaluating vaccine efficacy through anti SARS-CoV-2 reactivity in LMICs, specifically in sub-Saharan Africa. As limitation, this study did not evaluate the inter-test agreement between the two test kits used neither did it evaluate the rate of false-positive to RT-PCR that may have arisen following vaccination with the inactivated virus vaccine. The study did not also assess the titer of antibodies resulting from vaccination but rather evaluated the anti-SARS-CoV-2 reactivity through rapid antibody tests. In effect, absence of antibodies titration in the present study do not give way to clear correlation between the presence of anti-SARS-CoV-2 antibodies and effective protection against the infection. Nevertheless, the study revealed a lower risk of COVID-19 infection among vaccinated participants and even a higher prevalence of antibodies among those who received booster doses; suggesting a higher antibody titer/protection in these later ones. Unlike other studies in high income settings, assessing the titer and immunogenicity of these specific anti-SARS-CoV-2 antibodies through neutralization assays was not possible locally. However, serological antibodies tests employed in the current study captured antibodies against SARS-CoV-2 spike and nucleocapsid proteins, making them ideal choices for the detection of antibodies resulting from both natural infection and vaccination.

## Conclusion

Since its emergence, SARS-CoV-2 viral evolution and dynamics has been remarkable worldwide, with substantial impact both on individuals and healthcare systems. Despite the vaccine hesitancy observed at the beginning of the pandemic in several LMICs, the advent of COVID-19 vaccines was a turning point in the overall response against the disease. Our findings reveal a high prevalence of anti-SARS-CoV-2 antibodies, suggesting a certain degree of immunity/protection at community-level in Cameroon during the active circulation of omicron variants and sub-variants. Interestingly, vaccination with Pfizer and Moderna (i.e. mRNA vaccines) led to higher levels of anti-SARS-CoV-2 antibodies compared to other vaccines type. Even though further investigations may be needed for more in-depth characterization of these two vaccines (especially Pfizer), this result highlights a greater immunogenicity of mRNA-based vaccines in the current setting; advocating therefore for the prioritization of these vaccines for an effective protection in the advent of potential resurgence of SARS-CoV-2 infection. However, rapid antibody waning (~5months) calls for the need for vaccine updates in this tropical setting, especially with novel variants arising from a rapidly evolving virus and potentially compromising already acquired immunity.

## Acknowledgments

We want to acknowledge the Chantal Biya International Reference Centre (CIRCB) for hosting the present study and for all the supervision.

## Author contributions

**Conceptualization:** Desire Takou, Maria Mercedes Santoro, Francesca Ceccherini-Silberstein, Carlo-Federico Perno, Vittorio Colizzi, Nicaise Ndembi, Joseph Fokam.

**Data curation:** Ezechiel Ngoufack Jagni Semengue, Desire Takou, Alex Durand Nka, Naomi-Karell Etame, Joseph Fokam.

**Formal analysis:** Ezechiel Ngoufack Jagni Semengue, Sandrine Claire Ndjeyep Djupsa, Carla Montesano, Collins Ambes Chenwi, Grace Beloumou, Alex Durand Nka, Aurelie Minelle Kengni Ngueko, Evariste Molimbou, Naomi-Karell Etame, Davy-Hyacinthe Gouissi Anguechia, Audrey Rachel Mundo Nayang, Pamela Patricia Tueguem, Derrick

Tambe Ayuk Ngwese, Larissa Gaëlle Moko Fotso, Carlos Michel Tommo Tchouaket, Aude Christelle Ka'e, Nadine Fainguem, Cyrille Alain Abega Abega, Anne-Cecile Z-K Bissek, Claudia Alteri, Anne-Geneviève Marcelin, Carlo-Federico Perno.

**Funding acquisition:** Vittorio Colizzi.

**Investigation:** Ezechiel Ngoufack Jagni Semengue, Collins Ambes Chenwi, Grace Beloumou, Evariste Molimbou, Davy-Hyacinthe Gouissi Anguechia, Audrey Rachel Mundo Nayang, Pamela Patricia Tueguem, Therese Ndomgue, Derrick Tambe Ayuk Ngwese, Larissa Gaëlle Moko Fotso, Carlos Michel Tommo Tchouaket, Aude Christelle Ka'e, Nadine Fainguem, Cyrille Alain Abega Abega, Nadia Mandeng, Anne-Cecile Z-K Bissek.

**Methodology:** Ezechiel Ngoufack Jagni Semengue, Desire Takou, Marina Potesta', Collins Ambes Chenwi, Grace Beloumou, Alex Durand Nka, Aurelie Minelle Kengni Ngueko, Naomi-Karell Etame, Maria Mercedes Santoro, Claudia Alteri, Yap Boum, Francesca Ceccherini-Silberstein, Alexis Ndjolo, Carlo-Federico Perno, Joseph Fokam.

**Project administration:** Desire Takou, Carla Montesano, Alexis Ndjolo, Carlo-Federico Perno, Vittorio Colizzi, Nicaise Ndembi, Joseph Fokam.

**Resources:** Desire Takou, Evariste Molimbou, Pamela Patricia Tueguem, Therese Ndomgue, Nadine Fainguem, Nadia Mandeng, Emilienne Epee, Linda Esso, Claudia Alteri, Anne-Geneviève Marcelin, Francesca Ceccherini-Silberstein, Joseph Fokam.

**Software:** Ezechiel Ngoufack Jagni Semengue, Collins Ambes Chenwi, Alex Durand Nka, Aude Christelle Ka'e.

**Supervision:** Desire Takou, Emilienne Epee, Georges Etoundi Mballa, Carlo-Federico Perno, Jean Kaseya, Vittorio Colizzi, Nicaise Ndembi, Joseph Fokam.

**Validation:** Ezechiel Ngoufack Jagni Semengue, Desire Takou, Marina Potesta', Carla Montesano, Linda Esso, Georges Etoundi Mballa, Maria Mercedes Santoro, John Otokoye Otshudiema, Yap Boum, Anne-Geneviève Marcelin, Francesca Ceccherini-Silberstein, Alexis Ndjolo, Carlo-Federico Perno, Jean Kaseya, Vittorio Colizzi, Nicaise Ndembi, Joseph Fokam.

**Visualization:** Marina Potesta', Carla Montesano, Grace Beloumou, Evariste Molimbou, Naomi-Karell Etame, Emilienne Epee, Linda Esso, Maria Mercedes Santoro, Claudia Alteri, Yap Boum, Anne-Geneviève Marcelin, Francesca Ceccherini-Silberstein, Alexis Ndjolo, Carlo-Federico Perno, Jean Kaseya, Vittorio Colizzi, Nicaise Ndembi, Joseph Fokam.

**Writing – original draft:** Ezechiel Ngoufack Jagni Semengue, Desire Takou, Marina Potesta', Carla Montesano, Collins Ambes Chenwi, Grace Beloumou, Alex Durand Nka, Evariste Molimbou, Naomi-Karell Etame, Pamela Patricia Tueguem, Therese Ndomgue, Aude Christelle Ka'e, Nadine Fainguem, Nadia Mandeng, Emilienne Epee, Linda Esso, Maria Mercedes Santoro, Claudia Alteri, Yap Boum, Anne-Geneviève Marcelin, Francesca Ceccherini-Silberstein, Alexis Ndjolo, Carlo-Federico Perno, Jean Kaseya, Vittorio Colizzi, Nicaise Ndembi, Joseph Fokam.

**Writing – review & editing:** Ezechiel Ngoufack Jagni Semengue, Desire Takou, Marina Potesta', Carla Montesano, Collins Ambes Chenwi, Grace Beloumou, Alex Durand Nka, Evariste Molimbou, Naomi-Karell Etame, Pamela Patricia Tueguem, Therese Ndomgue, Aude Christelle Ka'e, Nadine Fainguem, Nadia Mandeng, Emilienne Epee, Linda Esso, Georges Etoundi Mballa, Maria Mercedes Santoro, Claudia Alteri, Yap Boum, Anne-Geneviève Marcelin, Francesca Ceccherini-Silberstein, Alexis Ndjolo, Carlo-Federico Perno, Jean Kaseya, Vittorio Colizzi, Nicaise Ndembi, Joseph Fokam.

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
