## [Decision Letter · Decision Letter 0]

18 Nov 2024

PGPH-D-24-02345

Disparities in anti-SARS-CoV-2 reactivity according to vaccines administered in the era of Omicron in Cameroon: lessons for future outbreak response

Dear Dr. Ngoufack Jagni Semengue,

Thank you for submitting your manuscript to PLOS Global Public Health. After careful consideration, we feel that it has merit but does not fully meet PLOS Global Public Health’s publication criteria as it currently stands. Therefore, we invite you to submit a revised version of the manuscript that addresses the points raised during the review process.

The reviewer comments have some overlap. Please respond to all of them but note if you do not think you need to make an edit in response to a question. Please note that reviewer 1 also included an attachment.

We look forward to receiving your revised manuscript.

Kind regards,

Abram L. Wagner, PhD, MPH

Academic Editor

Journal Requirements:

1. We noticed you have some minor occurrence of overlapping text with the following previous publication(s), which needs to be addressed:

-https://dx.doi.org/10.1128/spectrum.02732-21

-https://doi.org/10.1038/s41579-023-00878-2

In your revision ensure you cite all your sources (including your own works), and quote or rephrase any duplicated text outside the methods section. Further consideration is dependent on these concerns being addressed.

2. We do not publish any copyright or trademark symbols that usually accompany proprietary names, eg (R), (C), or TM  (e.g. next to drug or reagent names). Please remove all instances of trademark/copyright symbols throughout the text, including ® on page 18.

3. "In the online submission form, you indicated that "The datasets used and/or analyzed during the current study are available from the corresponding author on reasonable request.". 

a. In a public repository, 

b. Within the manuscript itself, or 

c. Uploaded as supplementary information.

Additional Editor Comments (if provided):

Reviewers' comments:

Reviewer's Responses to Questions

**Comments to the Author**

1. Does this manuscript meet PLOS Global Public Health’s publication criteria ? Is the manuscript technically sound, and do the data support the conclusions? The manuscript must describe methodologically and ethically rigorous research with conclusions that are appropriately drawn based on the data presented.

Reviewer #1: Yes

Reviewer #2: Yes

Reviewer #3: Partly

Reviewer #4: Yes

2. Has the statistical analysis been performed appropriately and rigorously?

Reviewer #1: Yes

Reviewer #2: Yes

Reviewer #3: No

Reviewer #4: Yes

3. Have the authors made all data underlying the findings in their manuscript fully available (please refer to the Data Availability Statement at the start of the manuscript PDF file)?

Reviewer #1: Yes

Reviewer #2: Yes

Reviewer #3: No

Reviewer #4: Yes

4. Is the manuscript presented in an intelligible fashion and written in standard English?

Reviewer #1: Yes

Reviewer #2: Yes

Reviewer #3: Yes

Reviewer #4: Yes

5. Review Comments to the Author

Reviewer #1: This manuscript examines the role of vaccines in combatting the COVID-19 pandemic in a low income country, specifically the levels of antibody response post infection and post vaccine and the rapid waning of antibody levels, indicating ongoing need for vaccine updates. The data collection, analysis and resulting conclusions are well considered and presented in the manuscript. I would recommend publication after addressing a number of recommended revisions to improve clarity.

Specific recommendations (Using the line numbering in the manuscript)

ABSTRACT:

Lines 45-46: The manuscript presents information on multiple vaccines, so I would recommend rewriting this sentence as follows: " With the advent of COVID-19, anti -SARS vaccines were a global health priority, but evidence on their significance with tropical settings remained limited."

Line 56: By "Global..." do you mean "National..."?

MAIN

Line 76: I would recommend replacing "..span.." with "...spread..." and replace "..propose.." with "...assess.."

Line 114: I would delete the word, "..unpreceded..."

Line 302: Again, I would recommend that you revise this sentence to refer to multiple vaccines rather than one. The revised sentence would read as follows, "For the past four years, anti-SARS-CoV-2 vaccines were considered as a global health priority, ..."

Lines 357-359. This sentence is unclear. Can the authors define the term "...translate...?

Lines 395: replace, "...compare.." with "..compared....."

Reviewer #2: General comment:

The study will further shape vaccine acceptance and indeed narrow down the types of vaccines that should be invested in and supplied to LMICs considering the huge costs for vaccination programming in our region; poverty, misinformation and hesitancy due to literacy levels and cultural considerations in rolling out vaccinations. I think the authors were smart in conceptualizing this study!!

Title: ok

Abstract:

Line 47- anti SARS-CoV-2...

Line 48- A community based cross-sectional....

Line 50- Plasma anti SARS-CoV-2 antibodies (both IgG and IgM) were tested....

Line 56- Global seroprevalence???

Line 57- Following logistic regression;......

Key words: ok

Introduction: ok

Materials and methods:

Line 132- This was a community based cross-sectional design in which....

Line 179- between both the tests was that.....

Line 207- Pearson's chi-square or Fisher's exact tests.....

Results: ok

Discussion: ok

Conclusion:

Line 395- compared to other....

Reviewer #3: The study is well justified; however, the study methods and design have several limitations in answering the study question and confirming the study conclusion.

1. Title: The title does not reflect the study's scope and design. Does the “Omicron Era” refer to the timing of vaccination administration, participant phlebotomy period or Omicron variant exposure?

2. Abstract: A description of “antibody waning” cannot be answered by this study design and laboratory analysis. Rapid tests only answer the presence and absence of antibodies; this cannot answer the decline of antibodies

3. Data Availability and Ethical Consideration:

It is unethical to collect personal identifiers such as names as part of socio-demographic data (sentence line 155). Given that the authors did not share this data until a reasonable request, this raises questions about data validity.

4. The timing of the local Omicron Circulation phase has not been cited (Sentence line 134).

5. The test kit's sensitivity and specificity have been validated after 20 days of symptom onset (sentence line 182); however, in this study, even symptomatic cases were included (sentence lines 152 & 224). The author needs to clarify this, including the exclusion criteria for this study

6. Two test kits have been used for antibody testing (sentence lines 168-169). Why has using two different kits not been justified, and if so, what was the inter-test agreement?

7. Please describe how the sample was handled for rt-RT PCR.

8. Were these kits specific for the Omicron variant?

9. A justification is needed for using a p-value of <0.2 for multivariate analysis. (sentence line 208)

10. What is the definition of partially vaccinated? Sentence line 246

11. Table 2's title is not specific to the table contents.

12. Several limitations to this study's generalization and reliability exist. Could the author include a section on these limitations?

Reviewer #4: This research on the protection offered by the COVID-19 vaccine is interesting. However, I believe the authors could improve the following points to better engage the reader:

Introduction:

The authors should mention the roles of antibodies in protection and outbreak control.

Materials and Methods:

1. Line 147: The phrase “participants fully complying with the study were finally included” should be clarified. Is this the selection criterion for this research? If so, please provide more details.

2. Line 151: The phrase “systematically screened” needs a more detailed description.

3. Line 190: The reference to the DaAn gene RT PCR kit (www.daangene.com) is incorrect. Please cite the exact kit used in this study.

4. RT-PCR Testing: RT-PCR was used to identify positive cases in this study. However, a small group of participants (4.8% as mentioned in line 244) received the Sinopharm vaccine, an inactivated virus vaccine. Evidence suggests that inactivated virus vaccines can significantly impact RT-PCR results by causing false positives. How do the authors plan to eliminate or identify false-positive RT-PCR results due to contamination from this vaccine?

Results:

1. Anti-SARS-CoV-2 Antibodies:

• Could the authors clarify if anti-SARS-CoV-2 antibodies are higher in the group that received both the infection and the vaccination compared to other groups?

• In Table 2a, could the authors explain why some groups (9 cases) exhibited symptoms but did not have anti-SARS-CoV-2 antibodies?

• In Table 2a, could the authors explain why the previously positive group (84 cases) lacked anti-SARS-CoV-2 antibodies? Did these individuals contract the virus naturally? Regarding vaccination and subsequent infection, what should readers understand about the loss of natural antibodies versus vaccine-induced antibodies? Please provide more details.

2. Carefully check the percentage ratios of all the results. Some totals exceed 100%.

Conclusion:

The authors mention that updated vaccines need to address emerging variants. However, it should be clearer and more precise whether new or current vaccines should be used based on the lessons learned from this study.

6. PLOS authors have the option to publish the peer review history of their article (what does this mean? ). If published, this will include your full peer review and any attached files.

**Do you want your identity to be public for this peer review?** For information about this choice, including consent withdrawal, please see our Privacy Policy .

Reviewer #1: **Yes: ** Paul Delay, MD, DTM&H (Lond)

Reviewer #2: No

Reviewer #3: No

Reviewer #4: **Yes: ** Diep The Tai

---

## [Editor Report · Decision Letter 1]

28 Nov 2024

PGPH-D-24-02345R1

Disparities in anti-SARS-CoV-2 reactivity according to vaccines administered in the era of Omicron in Cameroon: lessons for future outbreak response

Dear Dr. Ngoufack Jagni Semengue,

Thank you for submitting your manuscript to PLOS Global Public Health. After careful consideration, we feel that it has merit but does not fully meet PLOS Global Public Health’s publication criteria as it currently stands. Therefore, we invite you to submit a revised version of the manuscript that addresses the points raised during the review process.

We look forward to receiving your revised manuscript.

Kind regards,

Abram L. Wagner

Academic Editor

Journal Requirements:

1. We noticed you have some minor occurrence of overlapping text with the following previous publication(s), which needs to be addressed:

-https://dx.doi.org/10.1128/spectrum.02732-21

-https://doi.org/10.1038/s41579-023-00878-2

In your revision ensure you cite all your sources (including your own works), and quote or rephrase any duplicated text outside the methods section. Further consideration is dependent on these concerns being addressed.

Additional Editor Comments (if provided):

Thank you for submitting your manuscript to PLOS Global Public Health. I am the academic editor working on this manuscript. Before I send it out for peer review, I'd like you to edit your abstract a bit. I have put my comments below. I appreciate your willingness to turn in multiple rounds of manuscript edits before we send this out to peer review, and I think that making the abstract tighter will help me when I look for peer reviewers.

With the advent of COVID-19, anti-SARS-CoV-2 vaccine was a global health priority, but evidence on its significance within tropical settings remained limited. We sought to assess the disparity of anti-Sars-CoV-2 antibodies according to vaccine status and types of vaccines administered in Cameroon during Omicron waves.

- "disparity" here is a bit of a charged word. You could decide to keep it, but disparity typically refers to some difference that results from a historical injustice. I think you just mean "difference"?

A cross-sectional sero-survey was conducted from February-15 through July-31 2022 among individuals tested for COVID-19 in Yaoundé-Cameroon. Anti-SARS-CoV-2 antibodies (both IgG and IgM) were tested on plasma and statistical analyses were performed with p<0.05 statistically significant.

- generally it's okay to not mention the alpha level in an abstract. instead could you mention things like what kinds of statistical tests you did do? you mention logistic regression later but it's a bit confusing in its context.

Overall, 2449 participants were enrolled: median-age was 40 [31–49], 56.4% (1382/2449) men, 2.2% (54/2449) with flu-like symptoms and 19.6% (481/2449) reporting previous SARS-CoV-2 positivity. Regarding COVID-19 vaccination, 67.5% (1652/2449) had received at least one dose, 55.0% (909/1652) two-dose series and 37.1% (613/1652) received additional booster doses. Median duration from vaccination to phlebotomy was 5 [4–9] months. Global seroprevalence of anti-SARS-CoV-2 antibodies was 81.1% (1987/2449).

- recommend removing "global" here (could be misconstrued as "world wide")

Following logistic regressions, vaccine status, booster doses, post-vaccination time (≤5 months), Pfizer and Moderna vaccines, all remained associated with a high prevalence of anti-SARS-CoV-2 antibodies (aOR=1.95, aOR=1.36, aOR=1.64, aOR=2.07 and aOR=1.52 respectively; all p<0.05).

-  the first part of this phrase is confusing, since not sure if "vaccine status" is part of the list with "logistic regression", or if you mean "in a logistic regression, vaccine status (aOR=1.95), booster doses (etc.), ....,  were all associated with a high prevalence of antibodies.

This high seroprevalence of anti-SARS-CoV-2 antibodies suggests a certain degree of immunity/protection at community-level in Cameroon during Omicron waves, with Pfizer and Moderna inducing greater immunogenicity. However, rapid antibody waning (~5 months) calls for vaccine updates with novel variants (arising from a rapidly evolving virus) that could compromise already acquired immunity.
---

## [Decision Letter · Decision Letter 2]

3 Jan 2025

PGPH-D-24-02345R2

Disparities in anti-SARS-CoV-2 reactivity according to vaccines administered in the era of Omicron in Cameroon: lessons for future outbreak response

Dear Dr. Ngoufack Jagni Semengue,

Thank you for submitting your manuscript to PLOS Global Public Health. After careful consideration, we feel that it has merit but does not fully meet PLOS Global Public Health’s publication criteria as it currently stands. Therefore, we invite you to submit a revised version of the manuscript that addresses the points raised during the review process.

Please note that reviewer 4 and 7 have included their comments in an attachment.

We look forward to receiving your revised manuscript.

Kind regards,

Abram L. Wagner, PhD, MPH

Academic Editor

Journal Requirements:

Additional Editor Comments (if provided):

Reviewers' comments:

Reviewer's Responses to Questions

**Comments to the Author**

1. If the authors have adequately addressed your comments raised in a previous round of review and you feel that this manuscript is now acceptable for publication, you may indicate that here to bypass the “Comments to the Author” section, enter your conflict of interest statement in the “Confidential to Editor” section, and submit your "Accept" recommendation.

Reviewer #4: (No Response)

Reviewer #5: All comments have been addressed

Reviewer #6: All comments have been addressed

Reviewer #7: All comments have been addressed

Reviewer #8: All comments have been addressed

2. Does this manuscript meet PLOS Global Public Health’s publication criteria ? Is the manuscript technically sound, and do the data support the conclusions? The manuscript must describe methodologically and ethically rigorous research with conclusions that are appropriately drawn based on the data presented.

Reviewer #4: Yes

Reviewer #5: Yes

Reviewer #6: Yes

Reviewer #7: Yes

Reviewer #8: Yes

3. Has the statistical analysis been performed appropriately and rigorously?

Reviewer #4: Yes

Reviewer #5: Yes

Reviewer #6: Yes

Reviewer #7: Yes

Reviewer #8: Yes

4. Have the authors made all data underlying the findings in their manuscript fully available (please refer to the Data Availability Statement at the start of the manuscript PDF file)?

Reviewer #4: Yes

Reviewer #5: Yes

Reviewer #6: Yes

Reviewer #7: Yes

Reviewer #8: (No Response)

5. Is the manuscript presented in an intelligible fashion and written in standard English?

Reviewer #4: No

Reviewer #5: Yes

Reviewer #6: Yes

Reviewer #7: (No Response)

Reviewer #8: Yes

6. Review Comments to the Author

Reviewer #4: The authors of "Disparities in anti-SARS-CoV-2 reactivity according to vaccines administered in the era of Omicron in Cameroon: lessons for future outbreak response" demonstrated the effectiveness of different vaccines and their corresponding antibody responses during the Omicron wave. This study also highlights the protection provided by vaccines (approximately 5 months) and emphasizes the need for additional measures to protect the population. However, there are some biased data and unclear information in this research:

1. Why did the authors use two different antibody detection kits, such as Ninonasal and ABBEXA, when the ABBEXA kit covers the Ninonasal kit? How did the authors decide which test kit to use for each sample?

2. The study initially included 2,449 participants, but the authors only analyzed data from 1,987 participants in Table 2a. Could the authors clarify this discrepancy?

3. The authors did not provide the time interval between infection and vaccination. Did they have the vaccination schedule? If so, could they explain why the number of anti-SARS-CoV-2 antibodies decreased after the booster dose (538 vs. 752 in Table 2b)? Should we interpret this as indicating that booster doses are ineffective, or are there other reasons for this decrease? This relates to the conclusion about the effective duration of the vaccine. Do we need booster doses if they are ineffective?

4. The quality of the test kits is crucial for this study. However, the authors did not evaluate the inter-test agreement between the two kits, nor did they assess the rate of false positives compared to RT-PCR. How can the authors ensure the accuracy of their results? If the results are inaccurate, all conclusions will be meaningless. For example, in Table 2a, there were 1,590 cases of individuals with no previous positive results having anti-SARS-CoV-2 antibodies. Conversely, there were 84 cases of individuals without antibodies who had previous positive results. Can this be explained by antibody decay, vaccine ineffectiveness, or test kit quality? These data are unclear to me.

Reviewer #5: • Congratulations to the research team for the clarity of the objectives of the study and the strong statistical analysis conducted.

• I will suggest reorganizing the abstract to have a clear structure/format: Background, methodology, findings and conclusion for clarity and readability.

• I was unclear about the significance of the specific antibody types, IgG and IgM, and their relevance and whether the presence of IgG, IgM, or both was of any public health significance to the study results or their interpretation?

• Lines 233/234: Table 1, change the percentage on pregnant women from 1,2 to 1.2

• In the discussion line 384 and 385 made it clear that the study did not assess the titer of antibodies resulting from vaccination but rather evaluated the anti-SARS-CoV-2 reactivity through rapid antibody tests. Therefore, in the conclusion, the phrase in line 402, 403 and 404 ‘’ Interestingly, vaccination with Pfizer and Moderna (i.e. mRNA vaccines) led to higher levels of anti-SARS-CoV-2 antibodies compared to other vaccines type’’ appear confusing. Please rephrase for clarity indicating Pfizer and Moderna vaccines being associated with greater immunogenicity. Do ensure that it does not contradict the statement in the limitation of the study.

Reviewer #6: 1. In the abstract section the methodology part needs little more elaboration and the results part can be made short by omitting the irrelevant sentences.

2. The background is written properly.

3. The methodology is written properly in details regarding the laboratory procedure. However, the statistical analysis part is not properly written. These tests should be performed by any statistical software, but the authors have mentioned that t hey have done these tests by MS Excel. How? Please explain.

4. The Abs were measured by Lateral flow immunoassay. Do the authors evaluate the validity of this test? Please explain. Is these are the validated kits to detect the Abs?

5. Why PCR was performed on 3rd day? Please explain.

Reviewer #7: The manuscript genuinely provided information about the anti-SARSCOV-2 antibodies reactivity across all available kinds of COVID-19 vaccines administered during the era of Omicron in Cameroon. It is original and highlights the most potent and immunogenic vaccines administered during this era. I would like to recommend the manuscript for publication in the next PLOS Global Health Edition.

Reviewer #8: Dear authors, thank you very much for drafting this very important informative manuscript. I personally have a few minor comments for authors.

1. The abstract does not indicate the method used especially related to analysis.

2. Given the nature of the study design and the levels of anti-SARS-CoV-2 antibodies for each vaccine, I would like to see one of the recommendations being to further investigate each vaccine especially the Pfizer vaccine.

3. What are some of the potential biases that readers should consider when interpreting the findings of this study? I would like to see one of the findings looking at enrollment site

7. PLOS authors have the option to publish the peer review history of their article (what does this mean? ). If published, this will include your full peer review and any attached files.

**Do you want your identity to be public for this peer review?** For information about this choice, including consent withdrawal, please see our Privacy Policy .

Reviewer #4: **Yes: ** Diep The Tai

Reviewer #5: No

Reviewer #6: **Yes: ** Dr. Md. Abdullah Yusuf

Reviewer #7: No

Reviewer #8: No

---

## [Editor Report · Decision Letter 3]

5 Feb 2025

Disparities in anti-SARS-CoV-2 reactivity according to vaccines administered in the era of Omicron in Cameroon: lessons for future outbreak response

PGPH-D-24-02345R3

Dear Dr Ngoufack Jagni Semengue,

We are pleased to inform you that your manuscript 'Disparities in anti-SARS-CoV-2 reactivity according to vaccines administered in the era of Omicron in Cameroon: lessons for future outbreak response' has been provisionally accepted for publication in PLOS Global Public Health.

Best regards,

Abram L. Wagner, PhD, MPH

Academic Editor